# An Investigation of the Anisotropic Fatigue Properties of Laser Additively Manufactured Ti-6Al-4V under Vibration Loading

**DOI:** 10.3390/ma16145099

**Published:** 2023-07-19

**Authors:** Yan He, Wei Huang, Weiguo Guo, Yanping Li, Sihan Zhao, Dong Lin

**Affiliations:** School of Aeronautics, Northwestern Polytechnical University, Xi’an 710072, China; xauthy@mail.nwpu.edu.cn (Y.H.); yanping.li@mail.nwpu.edu.cn (Y.L.); zhaosihan@mail.nwpu.edu.cn (S.Z.); lindong@mail.nwpu.edu.cn (D.L.)

**Keywords:** laser additively manufactured Ti-6Al-4V, anisotropy, vibration fatigue, process-induced defects, microstructure

## Abstract

Laser additively manufactured (LAM) Ti-6Al-4V alloy has huge application potential in aerospace structural parts such as turbine blades. However, there are few studies on the fatigue properties of such LAM parts under vibration loading, particularly with regard to anisotropy. In this paper, vibration fatigue properties of LAM Ti-6Al-4V by laser melted deposition were investigated along the transversely deposited (TD) and parallelly deposited (PD) directions. Through the first-order bending vibration experiments, the LAM Ti-6Al-4V alloy exhibits obvious anisotropic fatigue properties and significant dispersion in fracture position. The fracture morphology analysis reveals that the vibration fatigue failure was mainly dominated by process-induced defects and microstructure. The fatigue strength at 10^6^ cycles of the samples with defect-free failure features (DFF) at initiation sites is 470.9 MPa in PD and 434.2 Mpa in TD, while that of the samples with defect-related failure features (DRF) at initiation sites is 364.2 Mpa in PD and 381.0 Mpa in TD. For the DFF group, the fatigue behavior is controlled by the prior β columnar grains with preferential orientation, which leads to enhanced fatigue crack propagation resistance for the PD samples. For the DRF group, which has lower fatigue lives, the fatigue anisotropy strongly depends on the projection area of the lack-of-fusion defects relative to the loading direction, resulting in better fatigue performance for the TD samples.

## 1. Introduction

Laser additive manufacturing (LAM) technology is becoming an increasingly popular manufacturing technology in the aerospace industry. In the LAM process, high-energy lasers deposit metallic wires or powders, layer by layer, to build components [1,2,3,4]. The high-efficiency and fast-forming process does not require subsequent expensive tooling and machining, enabling the direct fabrication of near-net-shaped components. Titanium alloys, for example Ti-6Al-4V, are probably the most widely studied LAM materials. Due to their good mechanical properties, especially their high performance in fatigue resistance and damage tolerance, titanium alloys are commonly utilized as turbine blades, compressor discs and blades, etc. Titanium alloy components always experience high cycle fatigue loading under vibration and resonance. This tends to induce crack initiation and propagation [5,6], which leads to fatigue failure and fracture [7,8]. Hence, revealing the vibration fatigue behavior of LAM-manufactured titanium alloy has great significance for ensuring the security of aerospace structures.

Unlike materials produced by traditional manufacturing processes, the final mechanical properties of LAM materials are influenced not only by processing parameters but also by solidification behavior. During the LAM process, short energy pulses produced highly localized melting and solidification, resulting in a strong temperature gradient [9]. The steep temperature gradients, combined with repetitive thermal cycling, caused macroscopically epitaxial growth in the grains. Moreover, the unique solidification behavior can inevitably lead to process-induced defects. It has been reported that spherical pores and lack of fusion (LoF) are the most common types of defects [10,11]. The LoF defects tend to extend in the scanning direction. Accordingly, the microstructure exhibits strong directionality, which consequently results in anisotropy of mechanical properties [12,13,14]. Lu et al. [15] and Carroll et al. [16] studied the tensile mechanical anisotropy of LAM Ti-6Al-4V, and the results showed that samples along the deposition direction present better ductility, while samples along the scanning direction exhibit a higher strength. Scholars suggested that the tensile anisotropy was caused by different deformation behaviors of the LoF defects and was aggravated by the different deformation mechanisms of the columnar grain boundaries [17,18]. Zhang et al. [19] studied the anisotropic tensile properties of selective electron beam melted (SEBM) Ti-6Al-4V and found the horizontally oriented samples had a lower yield strength and a higher ductility. The anisotropic mechanical properties are affected by the process-induced defects and the epitaxially elongated grains.

Recently, the fatigue properties of LAM-manufactured Ti-6Al-4V have received much attention. Most of these studies focused on traditional tensile fatigue behavior [20,21,22,23,24,25,26]. Bandyopadhyay et al. [20] studied the influence of building direction on the fatigue properties of laser-engineered net-shaped (LENS) Ti-6Al-4V and reported that the horizontally oriented samples showed a higher fatigue strength. This is due to the smaller defect size distribution on the cross-section of such oriented samples. Zhang et al. [16] found that the projected area of LoF defects vertical to the deposition direction is larger than that along the deposition direction, which results in lower fatigue strength in vertically oriented samples. Lv et al. [21] studied the anisotropic fatigue crack growth rate in laser-melted-deposited (LMD) Ti-6Al-4V and pointed out that horizontally oriented samples exhibited fatigue cracks interacting with only one or two columnar grains, resulting in lower fatigue crack growth resistance.

Structural components, such as blades, are prone to failure due to material fatigue when subjected to vibrational loading. Fatigue analysis in this context focuses on fatigue behavior near the natural frequency of flexible structures [27,28]. However, traditional fatigue experiments do not consider the natural frequency of the samples. Accordingly, traditional tensile fatigue data are insufficient to represent the failure of these components under vibration loading. Furthermore, traditional fatigue experiments performed via servo-hydraulic testing machines are known to be costly and time-consuming. Therefore, in recent years, researchers have investigated the fatigue behavior of LAM alloy based on first-order bending vibration tests using a shaker, which offers the advantage of shorter test durations [29,30,31]. Wanjara et al. [32] studied the fatigue performance of wire-fed electron beam additive-manufactured (WBD) Ti-6Al-4V thin-plated samples via vibration (resonance) fatigue testing, and calculated the fatigue limits and standard deviations of the samples. Zhao et al. and Ellyson et al. conducted vibration fatigue tests on Ti-6Al-4V samples fabricated by SLM [33], WBD [34], and LMD [35] processes, respectively. They found that process-induced defects were the primary factor contributing to lower fatigue lives and significant life dispersion. Zhao [35] examined the anisotropy of fatigue lives and crack propagation in LMD Ti-6Al-4V samples at a stress amplitude of 500 MPa and suggested that samples aligned with the deposition direction exhibited lower crack propagation speeds. However, a very limited amount of paper covered the anisotropic studies of the vibration fatigue behavior of the LAM Ti-6Al-4V alloy. This paper aims to address this gap in knowledge.

The purpose of this paper is to investigate the vibration fatigue properties of LAM Ti-6Al-4V samples manufactured using the LMD technology in both the transversely deposited (TD) and parallelly deposited (PD) directions. The paper is structured as follows. In Section 2, the microstructural characterization and fabrication of the samples are presented. The validity of the fatigue sample geometry and the feasibility of the fatigue experiment are verified. In Section 3, the fracture surface morphology, vibration fatigue failure mechanisms, and failure features at crack initiation sites are analyzed. Moreover, fatigue lives and fracture positions of the LAM samples in TD and PD based on failure features at initiation sites are analyzed. In Section 4, the influence of process-induced defects and columnar grain orientation on anisotropic fatigue behavior is discussed. Finally, a comprehensive conclusion is made in Section 5. By following this structure, the paper provides a detailed analysis of the vibration fatigue properties of LAM Ti-6Al-4V samples, including the examination of fracture surfaces, the analysis of failure features at initiation sites, and the influence of process-induced defects and columnar grain orientation on the anisotropy of vibration fatigue behavior. The paper contributes to the understanding of the fatigue performance of LAM Ti-6Al-4V and its potential applications in aerospace structural components.

## 2. Materials and Methods

### 2.1. LAM Process and Material

The material used in this paper was prepared using the LMD technology at the State Key Laboratory of Solidification Processing of Northwestern Polytechnical University (NPU; Xi’an, China). The Ti-6Al-4V alloy powder, with a size range of 44 to 178 μm, was dried at 120 °C under vacuum for 4 h to eliminate moisture absorption. The nominal chemical compositions (wt%) of the powder are presented in Table 1. Prior to deposition, the surface of the wrought Ti-6Al-4V substrate was cleaned using an acetone solution. The LAM manufacturing system (Bright Laser Technologies Co., Ltd., Xi’an, China ) utilized in this study consists of a 3D numerically controlled working table, a PRC4000 CO_2_ laser, a DPSF-2 automatic adjusted feeding system, and an inert gas chamber (oxygen content less than 50 ppm), as schematically shown in Figure 1. The scanning direction on each consecutive deposited layer was kept at 90°. The applied process parameters are listed in Table 2. The deposition direction was designated as the Z direction, while the two orthogonal scanning directions were designated as the X and Y directions, respectively. To alleviate residual stress within the Ti-6Al-4V alloy, the deposited part underwent a post-treatment procedure involving a solid solution and aging. This process included heating the part to 920 °C for 2 h, followed by air cooling (AC), and subsequently heating it to 520 °C for 4 h, followed by AC. The post-treatment not only helped mitigate residual stress but also enhanced the ductility and uniformity of the LAM alloy.

### 2.2. Microstructure Characteristics

To examine the microstructure of the LAM alloy, metallographic samples were prepared by stepwise polishing using sandpaper and subsequently etched for 60 s with Kroll’s reagent (1 mL HF: 3 mL HNO_3_: 50 mL H_2_O). The microstructure was then observed using optical microscopy (OM; OLYMPUS-GX71, Tokyo, Japan) and shown in Figure 2. On the X–Z plane, typical columnar β grains (390 ± 160 μm in width, several millimeters in length) grow epitaxially in the deposition direction and traverse several deposited layers. On the X–Y plane, however, a nearly equiaxed β microstructure can be observed (Figure 2d). The equiaxed grains were uniformly distributed, which can be attributed to the orthogonal scanning direction between consecutive layers (Figure 2). Therefore, the microstructural differences between the two scanning directions can be considered negligible. At β grain boundaries (GBs), grain boundary α phases (1–3 μm in width) were clearly identified, as in Figure 2c,e. On either side of the GBs, parallel needle-like martensitic α grains constituted α colonies (60–90 μm in width). The colonies could extend up to a few hundred microns or even 1 mm along the deposition direction. However, along the scanning direction, the directional consistency of the colonies decreased due to the presence of equiaxed grains. The microstructure analysis reveals the presence of distinct columnar β grains in the deposition direction and equiaxed β grains in the orthogonal scanning direction. The presence of grain boundary α phases and martensitic α colonies further characterizes the microstructure of the LAM alloy.

Inevitably, process-induced defects are formed in the LAM Ti-6Al-4V alloy, as shown in Figure 3. The defects can be classified into circular voids with smooth edges (Figure 3a) and LoF defects exhibiting irregular geometrical morphologies with sharp corners (Figure 3b). To visualize and quantify the porosity within the LAM alloy, a sample measuring 3 × 9 × 9 mm^3^ was cut from the center of the alloy. X-ray computed tomography (XCT; MU2000-D, YXLON Ltd., Hamburg, Germany) was employed, operating at a voltage of 205 kV and a current of 200 μA, with a resolution of 10 μm. The XCT scanning generated 2D radiographs, which were reconstructed into a 3D visualization of voids using Dragonfly software, as in Figure 3c. The porosity was measured to be approximately 0.006% with a random void distribution. In Figure 3d,e, the 3D representation of the two types of defects correlated well with the observations made in the optical images. 

### 2.3. Samples

To investigate the distinct fatigue behavior in TD and PD, two types of samples were machined by wire electrical discharge machining. These samples were categorized as TD samples and PD samples, as schematically shown in Figure 4a.

The first step in vibration fatigue experiments was to determine the geometry of the fatigue sample. As reported by George et al. [27], a continuous system has an infinite number of vibration modes, each corresponding to a specific natural frequency. The first-order vibration mode is commonly observed in moving parts of power engineering systems, such as blades. Additionally, the first-order vibration mode is more feasible to replicate in fatigue experiments compared to the higher-order vibration modes [28,29,30,31,32,33,34,35]. Accordingly, the first-order vibration mode shape was chosen as the target mode. In this study, the geometries of the vibration fatigue sample were obtained through finite element simulation using an iterative method. To ensure that the fatigue crack initiates near the notch root, a round notch with a radius of 8.5 mm was processed on both sides of the sample (Figure 4b). A through-thickness hole with a diameter of 5 mm was designed to enable the cantilever sample to be connected to an electromagnetic shaker. Another through-thickness hole with a diameter of 3 mm was utilized to attach a mass block (YG8, 14.44 × 9.2 mm^2^, 21.2 g, Figure 5). To minimize the influence of surface roughness on fatigue behavior, the gauge section surfaces of the samples were sanded with #180–#5000 sandpaper and polished with diamond suspension. The measured Ra values ranged from 0.4 to 0.6 μm, indicating a relatively small variation in surface roughness. Therefore, the effect of surface roughness on fatigue properties can be neglected in this study.

### 2.4. Vibration Fatigue Experiments

Vibration fatigue experiments were performed on a JZK-200 electromagnetic shaker (Figure 5) (YuTian Co., Ltd., Wuxi, China). The parameters of the shaker are listed in Table 3. The equipment used in the test system consisted of the shaker, a YE1311 signal generator and a YE5873A power amplifier. The instrumentation used in the measurement system is a LKGD500 displacement sensor, a SDY2301 strain gauge bridge box, and BE120-03AA ultrasonic dynamic strain gauges. During experiments, the cantilever sample was clamped in a cylindrical fixture through the left through hole, ensuring consistent compression at the same position for each sample. The sample tip displacement was monitored by the displacement sensor. The laser target was positioned 4.7 mm away from the center of the mass block. The stress was monitored by the strain gauges glued longitudinally on the surface of the samples, exactly located at the maximum stress position. All fatigue experiments were conducted under laboratory conditions, with the samples excited to vibrate at the first-order natural frequency in full reversed condition (stress ratio R = −1).

To assess the feasibility of the cantilever sample and provide reference data for frequency regulation and vibration stress calibration during experiments, a modal analysis was performed using ABAQUS software (Version 6.14, Dassault Systèmes Ltd., Paris, France). The modal shape, first-order natural frequency, and vibration nodal region were analyzed. The model consists of two parts, the cantilever sample and the mass block. Material parameters from the traditional Ti-6Al-4V alloy were applied, as listed in Table 4. The displacement and rotation of the clamped edge of the sample were completely constrained, which was the same as that adopted in the experiments. As the mesh sensitivity affects the magnitude and distribution of the stress state near the notch root, the mesh was refined in the notch region of the sample, as shown in Figure 6a. The mesh size at the notch region is about 50 μm. The notch region consists of 6 elemental layers along the thickness of the sample. As a result, there are 124,676 elements and 198,376 nodes included in the model, as listed in Table 4. In addition, we ignored the anisotropy of material properties in the model. This is because the purpose of the modal analysis was only to provide reference data for the frequency and vibration stress during experiments. The computed first-order natural frequency was 47.4 Hz, while the measured frequency ranged from 46 to 50 Hz during frequency sweeping for the tested LAM samples. This indicates that the error between the measured and simulated natural frequencies is less than 5%, affirming the accuracy of the model analysis.

Figure 6b shows the first-order modal displacement distribution on the sample surface. It can be observed that the displacement gradually decreases from the sample tip to the clamped edge, with the maximum vibration displacement occurring at the sample tip. The modal stress distribution is presented in Figure 6c. The stress is significantly higher on the surface of the notch region compared to other regions, with the maximum vibration stress observed on this surface. The notch region is considered the fatigue region, representing the theoretical weak position and predicted fracture position. Additionally, a continuous stress distribution curve along the central path on the surface near the notch region is obtained (Figure 6d). The origin is set at the minimum cross-section of the fatigue region. Notably, the stress distribution remains consistent at different tip displacements. The maximum stress was not found right at the minimum cross-section; it slightly drifted at the point of −0.308 mm. To validate the simulated results, strain gauges were attached to the maximum stress position. The comparison of maximum stress from the simulated results and strain gauge measurements at different tip displacements is shown in Figure 6d,e, revealing excellent consistency. Therefore, the ABAQUS results provide a reliable reference for determining the location of the fatigue region in the present sample.

Furthermore, it can be clearly observed that the maximum stress amplitude is linear with the sample tip displacement (Figure 6e), which has also been reported in previous vibration fatigue experiments [33,34,35]. This implies that the maximum stress amplitude can be controlled by adjusting the sample tip displacement. In this paper, stress-controlled fatigue experiments were conducted within a maximum stress amplitude range of 300–600 MPa, which corresponds to approximately 55% of the yield strength as mentioned in our previous study [35]. During experiments, when a fatigue crack was initiated in the surface layer of the cantilever samples, the natural frequency of the samples would shift [32,33]. Once the natural frequency decreased by 2%, the excitation frequency was adjusted to match the real-time natural frequency, thereby accelerating the failure of the samples. A total of nineteen TD samples and sixteen PD samples were used for the fatigue experiments. The fracture surfaces after experiments were observed using a scanning electron microscope (SEM; JEOL JSM-6390A, Tokyo, Japan).

## 3. Results and Analysis

Figure 7 displays a representative fatigue fracture surface from the TD group at a maximum stress amplitude of 500 MPa (N_f_ = 32.88 × 10^4^ cycles). The force analysis of the cantilever sample is shown in Figure 7a. The sample was subjected to cyclic tensile or compressive stress under vibration loading. The maximum tensile stress occurs at the two side surfaces of the fatigue region (notch root), which is significantly higher than that within the interior of the sample. This observation is consistent with the stress distribution on the cross-section of the notch root, as shown in Figure 7b′. The fracture surface can generally be distinguished into three main zones: the crack initiation zone, the crack propagation zone, and the fast fracture zone (Figure 7b). Figure 7c–e shows the crack initiation sites. A spherical void is observed at the initiation site in the lower surface (Figure 7c), and several crack initiation sites caused by surface slip are arranged in the upper surface, as in Figure 7d,e. These indicate that fatigue damage occurs at the two side surfaces of the notch region, leading to crack initiation. The crack then propagates further inward until the sample ultimately fails.

In Figure 7b, the crack propagation zone area corresponding to the initiation zones on the upper and lower surfaces differs significantly. The area on the lower surface is larger, while that on the upper surface is extremely small. This discrepancy is attributed to the fact that the surface defects in the lower surface may preferentially generate local micro-stresses greater than the yield strength value under vibration loading. Accordingly, the crack propagation rate is higher on the lower surface due to the larger tensile stress. Consequently, the crack propagation area on the lower surface is larger than that on the upper surface. Thus, the void in the lower surface serves as the primary crack origin and the main controlling factor affecting fatigue life. Throughout this paper, only the failure features at the main initiation sites will be discussed. At higher magnification, typical fatigue striations and secondary cracks can be clearly observed in Figure 7f. The fatigue striations were formed by the opening–closing actions of cracks due to the high stress at the crack tip [36]. Trans-granular crack growth across the α grains can be seen in Figure 7g. Figure 7h shows the fast fracture zone located between the crack propagation zones. In this zone, there are numerous small and shallow dimples, which were generated by the micro-pores and their tearing ridges.

All fatigue experimental results in the TD and PD are summarized in Table 5. The reference maximum stress (σ_ref-max_) was calculated by the linear relation between the maximum stress amplitude and sample tip displacement (A). Representative fatigue failure features at crack initiation sites are shown in Figure 8. It can be seen that there are two kinds of fatigue failure features at initiation sites: defect-related failure (DRF) and defect-free failure (DFF). For the DRF group, process-induced voids or LoF defects were found at crack initiation sites, as in Figure 8a,b,d,e. While for the DFF group, no defects were found at crack initiation sites, as in Figure 8c,f, and microstructural facets proved to be the typical failure features. This type of failure is commonly observed in fully lamellar α + β titanium alloys, such as Ti-6Al-4V alloy [37,38,39]. In LAM Ti-6Al-4V alloy, α colonies are the most typical characteristic microstructure. Individual α grains within a single colony are aligned perfectly, enabling the sharing of the same slip system. This allows for relatively unimpeded slip propagation across an entire colony. Additionally, α colonies are considered to be the microstructural features presenting the longest slip length in the material [13], resulting in a shorter crack initiation time. As such, the microstructural facets can be considered sheared α colonies. 

Previous studies have demonstrated that fatigue fracture position can vary on a microscopic scale [35,40]. In this paper, the fracture positions of the LAM samples in the TD and PD were summarized and shown in Figure 9b,c. The data were further distinctively identified and grouped according to the observed failure features at crack initiation sites. The modal stress distribution in the fatigue region is also shown in Figure 9a. The black dashed line represents the minimum cross-section, which is defined as the origin of the vertical axis. The positive axis is from the clamp edge to the sample tip edge. The red double-dot dash line represents the position of the maximum stress, i.e., the theoretical weak section. From Figure 9b,c, it can be observed that for both oriented samples, the fracture position exhibits significant variation between the DRF and DFF groups. Specifically, a large scatter in the fracture position is observed for the DRF group; most of the position is distributed between −2.4 and 2.6 mm in TD and between −2.4 and 1.9 mm in PD. However, the scatter appears to decrease for the DFF group; most of the fracture position is distributed between −1 and 0.7 mm. These findings indicate that defects contribute to the local stress field, thereby influencing fatigue failure and resulting in a drift in the fracture position.

Considering that a significant number of fracture positions deviate from the maximum stress position, there may be large differences between the true maximum stress (σ_f-max_) at the fracture position and the reference maximum stress (σ_ref-max_). The farther the fracture position is from the maximum stress position, the lower the stress at the fracture position. Therefore, a modification of σ_ref-max_ is a prerequisite to obtaining σ_f-max_. σ_f-max_ was calculated based on the relation between the continuous stress distribution curves and the fracture position.

Figure 10 shows the fatigue S–N data in terms of the σ_f-max_ value versus N_f_. The PD samples are denoted by a square symbol, while the TD samples are denoted by a triangle symbol. It is evident that the vibration fatigue lives exhibit significant dispersion, with the scatter of fatigue lives spanning two orders of magnitude, particularly at low-level stress. The PD samples demonstrate a more dispersed fatigue life compared to the TD samples. A linear fit was performed on the S–N data in the TD and PD, as indicated by the blue solid and dashed lines in the figure. Clearly, the fatigue properties in the TD and PD are similar, indicating that the influence of sample orientation on fatigue lives is not easily distinguishable without considering the fatigue failure features.

The data for the DRF and DFF are distinguished by different-colored legends, as in Figure 10. Four red and black lines were obtained by a linear fit of the S–N data for each set of data. It is evident that the fatigue lives of the DFF group, both in TD and PD, are generally higher than those of the DRF group under the same σ_f-max_. This difference in fatigue life can be attributed to the high sensitivity of fatigue performance to defects, as these defects reduced the crack resistance under vibration loading. Meanwhile, vibration fatigue behavior for both the DRF and DFF groups exhibits apparent anisotropy. For the DFF group, the PD samples show better fatigue performance compared to the TD ones. However, for the DRF group, the TD samples exhibit higher fatigue lives than the PD ones under high-level stress. In order to quantify and compare the fatigue strength of samples in the four different groups, the calculated fatigue strength based on the linear fit is listed in Table 6. It can be seen that the samples of the DFF group in PD possessed a higher fatigue strength (470.9 MPa) at 10^6^ cycles as compared to those in TD (434.2 MPa). The samples of the DRF group in PD possessed a lower fatigue strength (364.2 MPa) as compared to those in TD (381.0 MPa). Notably, the fatigue lives of the TD and PD samples show little difference when the applied stress is relatively low (less than 350 MPa). 

## 4. Discussion

### 4.1. Effect of Columnar Grains Orientation on the Anisotropic Fatigue

It was previously found that there are three main factors affecting the anisotropic fatigue behavior of LAM materials: residual stress, process-induced defects, and directional microstructure. Considerable residual tensile stress is often present on the surface of deposited Ti-6Al-4V alloy along the deposition direction [41], and it can reduce the fatigue resistance of PD samples. However, in this paper, the residual stress has been eliminated through the subsequent solution and aging treatment, as mentioned in 2.1. Thus, residual stress is unlikely to be the primary cause of the observed anisotropic vibration fatigue properties. 

As for the DFF group, fatigue performance of the PD samples is found to be superior to the TD ones, which is consistent with what is reported in [26,42,43]. Since no significant defects were observed at crack initiation sites, defects are not considered to be the main cause of the anisotropy. Instead, the anisotropic fatigue performance is more likely related to the directional microstructure. Figure 11 shows representative fracture surfaces from the DFF group in two directions, exhibiting distinct characteristic features. In the TD sample, multiple small slopes alternate in the fatigue crack propagation zone along the sample width direction, i.e., the scanning direction. The elongated features (parallel to the hollow arrows) resemble the shape of columnar grains (Figure 11a). The width and the indicated orientation of these features agree well with the microstructural characterization presented in the X–Z plane (Figure 2). At higher magnification (Figure 11b), different appearances of the fatigue crack propagation paths can be observed. Parallel α-laths are commonly found in region I, which appears smoother compared to region II. Additionally, a few staircase-like features are present along the fatigue crack propagation path in region II. The different appearance could be attributed to the different slip characteristics in different regions. This may be because both sides of the β boundaries belong to different β grains. In the PD samples, the fracture surfaces exhibit pronounced or concave morphologies, as indicated by the ellipses (Figure 11c–e). These features align with the equiaxed grains presented in the X–Y plane (Figure 2). These features may be related to the fact that the fracture surfaces of the PD samples belong to the X-Y plane. 

Figure 12 illustrates a schematic diagram indicating the fatigue crack propagation direction with respect to the columnar grains or deposition direction. In LAM, columnar grains extend along the deposition direction and exhibit elongated features. The density of GBs along the scanning direction is significantly higher compared to along the deposition direction. For the TD sample, fatigue cracks typically propagate along the deposition direction and encounter the lower-density GBs. Conversely, for the PD sample, fatigue cracks propagate perpendicular to the deposition direction and encounter the higher-density GBs. The density of GBs is positively correlated with crack growth resistance. Therefore, less resistance occurs on the crack propagation path for the TD sample, and therefore, they experience less deflection in the path. However, a higher crack growth resistance can be expected for the PD sample (Figure 12b). The obstruction of fatigue crack propagation due to the higher-density GBs is the likely cause of the higher fatigue performance of the PD sample. Additionally, the presence of β grain boundary α colonies is also considered to contribute to the anisotropy of the material’s fatigue property. When crack propagation encounters GBs, it interacts with the α colonies. Xie et al. [44,45] demonstrated that the colonies have higher fatigue crack propagation resistance compared to basket-weave structures. Therefore, the high density of grain boundary α colonies in the PD also contributes to the higher crack propagation resistance.

### 4.2. Effect of Process-Induced Defects

As for the DRF group, at high-level stress (from 350 MPa to 600 MPa), the vibration fatigue performance of TD samples is superior to that of PD samples, consistent with previous reports [25,46]. However, at low-level stress (less than 350 MPa), the level of anisotropy in fatigue performance decreases with the increase in the number of cycles (Figure 10). These indicate that both defects within the LAM material and applied stresses may affect the vibration fatigue lives of the DRF group. Representative fracture surfaces of the DRF group in the two directions fatigued at similar stress levels of 500 MPa and 370 MPa are examined, as in Figure 8. The defect failure features, especially the LoF defects with sharp angles, at crack initiation sites for the TD and PD samples are different. For the TD samples, besides a few voids caused by entrapped gas during manufacturing, LoF defects can be observed with a narrow appearance on the X–Z plane (Figure 8a). However, for the PD samples, the LoF defects can cover a broad cross-sectional region on the X–Y plane (Figure 8d,e). This is just attributed to the fact that the loading direction is perpendicular to the elongated LoF defects for the PD samples and along the LoF defects for the TD ones. The LoF defects, which cover a broad cross-sectional region on the X–Y plane, result in a larger reduction in load-bearing area. On the contrary, narrow LoF defects, which were generally obtained for the TD samples, exhibit a smaller reduction in load-bearing area. The larger reduction in load-bearing area for PD samples may result in an increase in the actual loading force. The sharp angles of the LoF defects could result in local stress concentrations and lead to early fracture [47]. Hence, both the reduced load-bearing area and the increased level of stress concentration reduce fatigue, particularly in PD samples. Furthermore, after approximately ~10^5^ cycles, the reduced loading force results in a decrease in stress concentration around the defects, reducing the difference in fatigue response between the two oriented samples.

## 5. Conclusions and Future Outlook

In this paper, first-order bending vibration fatigue properties of LAM Ti-6Al-4V cantilever samples in different directions (PD and TD) were investigated. The vibration fracture morphology was observed, and the fracture position and fatigue life were evaluated. The difference in fatigue performance can be totally accounted for by the different failure mechanisms shown to be present in LAM material, i.e., process-induced defects and α colonies. The key findings can be summarized as follows:

(1) Process-induced defects (voids or lack-of-fusion defects) and α colonies located at or near the sample surfaces play a significant role in fatigue failure initiation. The samples with defect-related failure features (DRF) at initiation sites exhibit inferior fatigue performance compared to the samples with defect-free failure features (DFF).

(2) The fracture position varies depending on the failure features at crack initiation sites. For the DRF group, the fracture position shows a large dispersion and is not necessarily localized at the maximum stress position. However, for the DFF group, the fracture position tends to be shifted towards the minimum cross-section.

(3) The vibration fatigue properties of both the DRF and DFF groups show apparent anisotropy. For the DFF group, the PD samples exhibit better fatigue performance. This anisotropy is mainly controlled by the prior β columnar grains. The different obstruction of fatigue crack propagation due to the density of the grain boundaries in the two directions results in anisotropic fatigue resistance, while the different fatigue crack resistance of the β grain boundary α colonies is believed to aggravate the anisotropy.

(4) In contrast, for the DRF group, the TD samples exhibit higher fatigue lives compared to the PD ones. This anisotropy is dominantly dependent on the LoF defects. The LoF defects present irregular shapes, with the longest axis distributing predominantly vertically in the deposition direction. It leads to a larger defect area and a higher stress concentration in the PD samples than in the TD samples, which significantly reduces fatigue resistance. However, at low levels of stress, the level of anisotropy in fatigue performance decreases with the number of cycles.

This study proves that the initiation and propagation of vibration fatigue cracks in AM materials are closely related to the process-induced defects and microstructure (columnar grains or α colonies). The results and discussion provide a basis for further regulating the vibration fatigue lives of AM materials. For AM, anisotropy’s contribution to fatigue performance needs to be better understood, as well as factors not considered here such as surface finish and porosity spatial distribution. This points out the directions for the authors’ future research.

## Figures and Tables

**Figure 1 materials-16-05099-f001:**
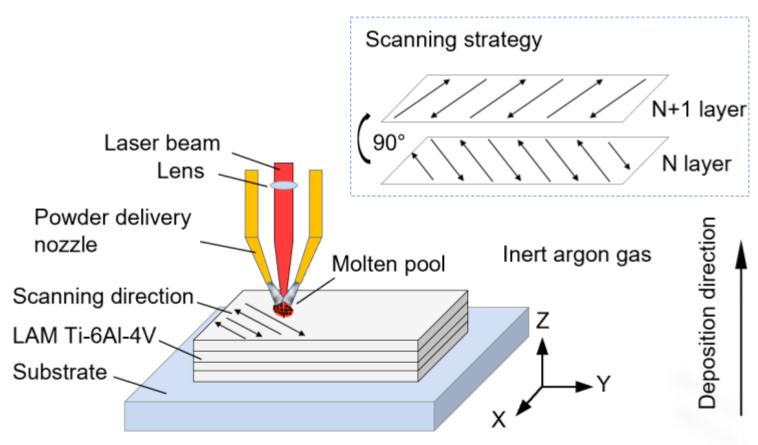
Schematic of LAM process.

**Figure 2 materials-16-05099-f002:**
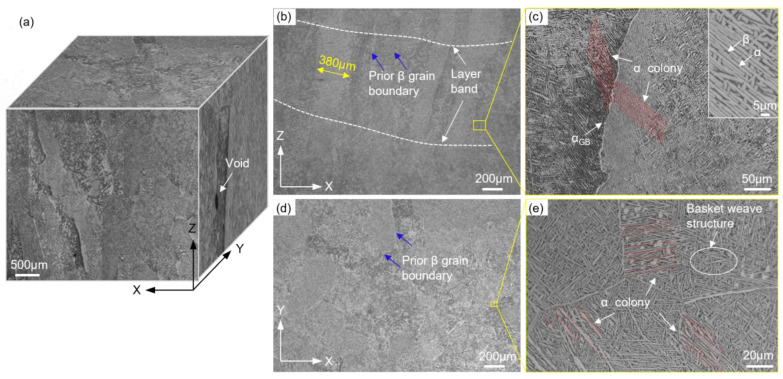
(**a**) Three-dimensional metallography of the LAM Ti-6Al-4V. Microstructure in (**b**) deposition direction and (**d**) scanning direction. (**c**,**e**) The magnification of the selected area in (**b**,**d**), respectively. The white phase is α phase, and the black phase is β phase. The blue arrows represent prior β grain boundaries, and the red lines represent α colonies.

**Figure 3 materials-16-05099-f003:**
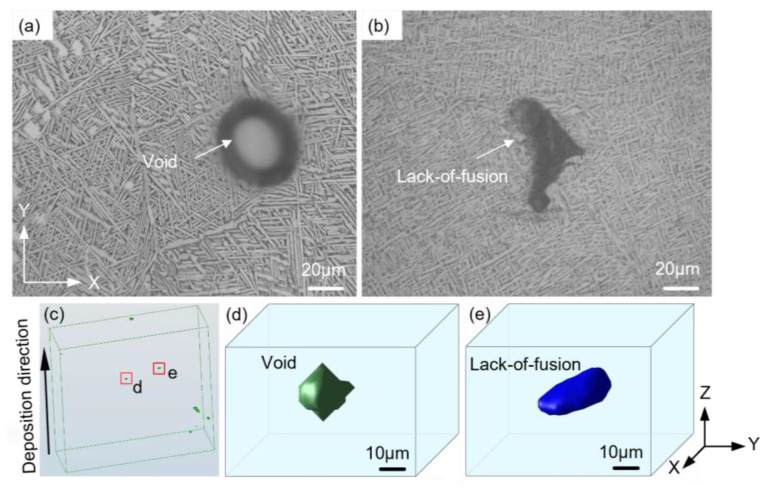
Representative defects in LAM Ti-6Al-4V: (**a**) circular void, (**b**) irregular LoF defect. (**c**) X-ray computed tomography scan results. (**d**,**e**) The magnification of the selected area in (**c**).

**Figure 4 materials-16-05099-f004:**
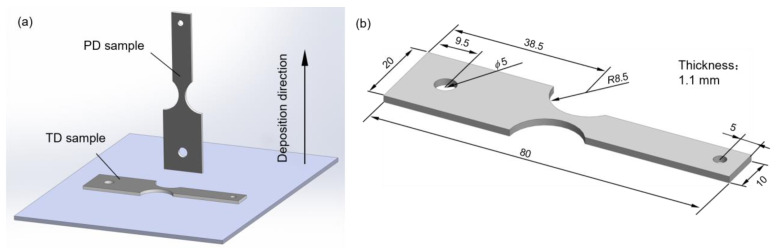
(**a**) Schematic of the tested samples in different orientations. Geometries (unit: mm) of (**b**) vibration fatigue sample.

**Figure 5 materials-16-05099-f005:**
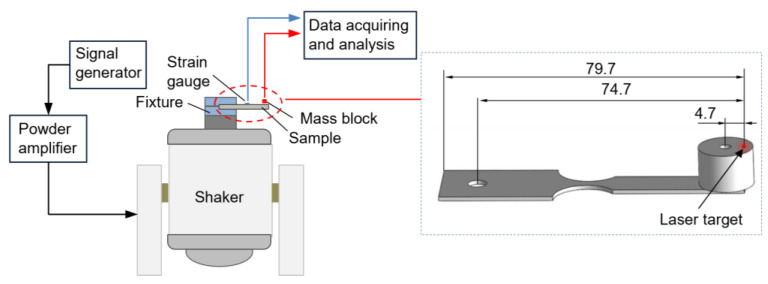
Vibration fatigue experimental setup.

**Figure 6 materials-16-05099-f006:**
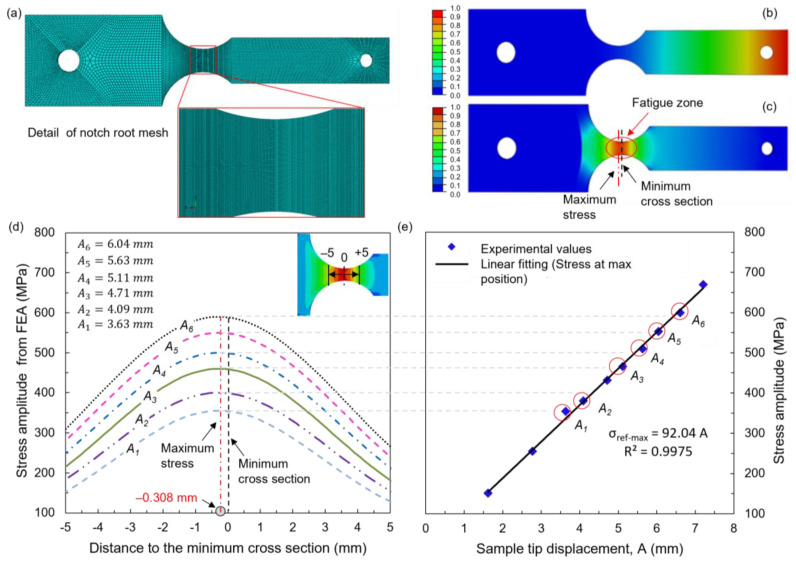
Modal analysis of Ti-6Al-4V sample by ABAQUS: (**a**) detail of the meshing, (**b**) displacement distribution, (**c**) stress distribution, and (**d**) stress distribution curve along the central path at different tip displacements. (**e**) Fitting curve for the maximum stress amplitude and sample tip displacement.

**Figure 7 materials-16-05099-f007:**
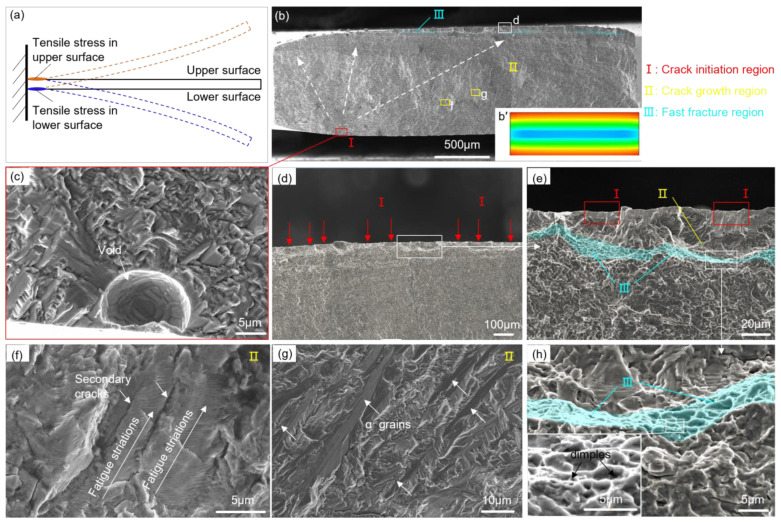
(**a**) Force analysis of cantilever sample. Representative fatigue fracture surfaces: (**b**) the macroscopic fracture; (**b′**) stress distribution on the cross-section of the notch root; (**c**–**e**) the crack initiation site; (**f**,**g**) the crack propagation zone; and (**h**) the fast fracture zone.

**Figure 8 materials-16-05099-f008:**
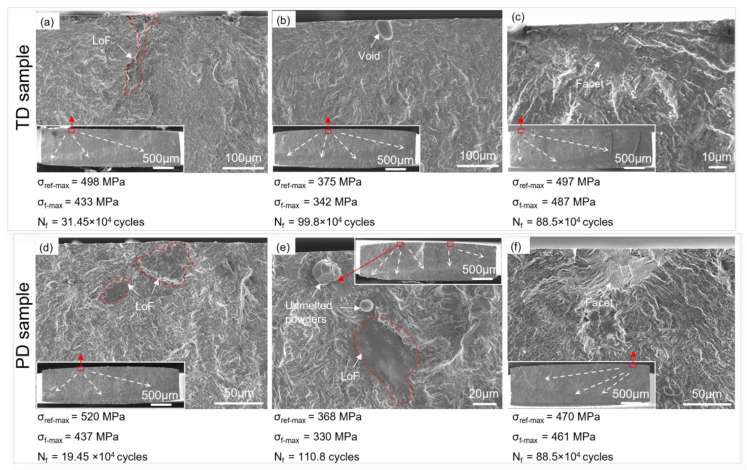
Comparison of fatigue failure features at crack initiation sites in TD and PD: (**a**,**b**,**d**,**e**) DRF, and (**c**,**f**) DFF. The red rectangles represent crack initiation sites. The white dotted lines with arrows represent crack propagation direction. Note: σ_ref-max_-reference maximum stress, σ_f-max_-true maximum stress at fracture position.

**Figure 9 materials-16-05099-f009:**
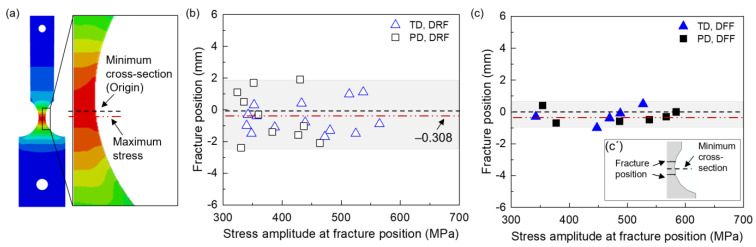
(**a**) Local stress distribution from ABAQUS. Fracture position of (**b**) DRF group, and (**c**) DFF group. (**c´**) Schematic of fracture position.

**Figure 10 materials-16-05099-f010:**
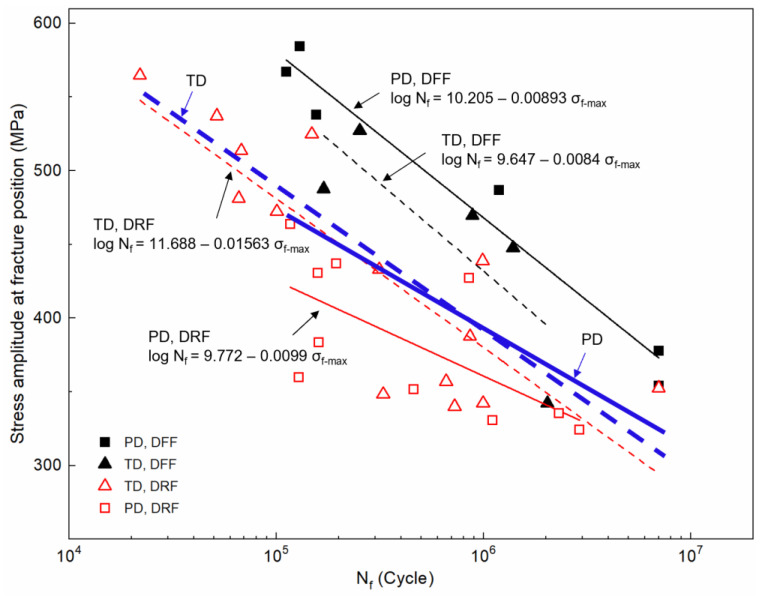
S–N data (including the equations for linear fit) of LAM samples. The stress values were presented as the true stress at fracture position.

**Figure 11 materials-16-05099-f011:**
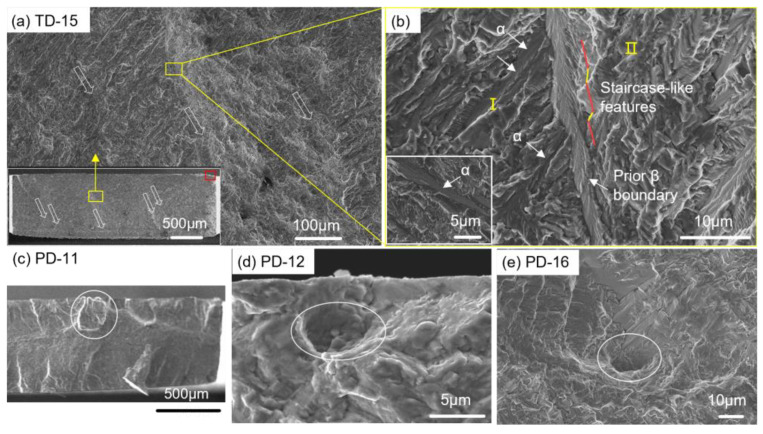
Representative fracture surfaces of the DFF group from (**a**) TD and (**c**–**e**) PD samples. (**b**) The magnification of the selected area in (**a**). The red rectangle and hollow arrows in (**a**) represent crack initiation site and β grain orientation, respectively. Region I. and Region II. in (**b**) represent the regions on either side of the β grain, respectively.

**Figure 12 materials-16-05099-f012:**
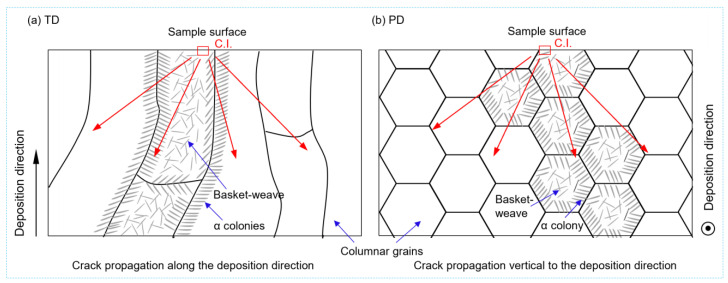
Schematics indicating crack propagation along and vertical to the columnar grains: (**a**) TD sample; (**b**) PD sample. C.I. represents the crack initiation site.

**Table 1 materials-16-05099-t001:** Chemical compositions of the Ti-6Al-4V powders (wt%).

Element	Al	V	Fe	Si	C	N	H	O	Ti
Mass fraction	6.02	4.00	0.098	0.033	0.025	0.04	0.008	0.16	Bal.

**Table 2 materials-16-05099-t002:** LAM processing parameters.

Laser Power, kW	Scanning Velocity, mm/min	Powder Feeding Rate, g/min	Overlap Ratio, %
7	600–900	40–60	30–50

**Table 3 materials-16-05099-t003:** Parameters of the shaker.

Frequency	Maximum Excitation Force	Maximum Acceleration	Excitation Power	Maximum Peak to Peak Displacement
10–500 Hz	2 kN	45 g ^1^	2.2 kW	±20 mm

^1^ Gravitational acceleration.

**Table 4 materials-16-05099-t004:** The material and model parameters in ABAQUS analysis.

Young Modulus	Poisson Ratio	Density	Element Type	Number of Elements	Number of Nodes
116 GPa	0.27	2730 kg/m^3^	C3D8R	124,676	198,376

**Table 5 materials-16-05099-t005:** Information about fatigue experiments results and observed failure features at crack initiation sites.

Sample	A, mm	σ_ref-max_, MPa	σ_f-max_, MPa	N_f_, ×10^4^	Failure Feature	Sample	A, mm	σ_ref-max_, MPa	σ_f-max_, MPa	N_f_, ×10^4^	Failure Feature
TD-1	5.78	580	565	2.20	DRF ^1^	PD-1	4.75	504	464	11.70	DRF
TD-2	5.50	580	537	5.17	DRF	PD-2	4.47	450	427	19.45	DRF
TD-3	5.25	550	514	6.77	DRF	PD-3	4.40	500	430	15.90	DRF
TD-4	4.92	500	481	6.59	DRF	PD-4	4.37	450	427	85.45	DRF
TD-5	4.83	500	472	10.09	DRF	PD-5	3.92	400	383	16.05	DRF
TD-6	4.80	550	525	14.86	DRF	PD-6	3.68	368	360	12.88	DRF
TD-7	4.49	450	433	99.28	DRF	PD-7	3.60	400	352	46.05	DRF
TD-8	4.43	450	439	31.45	DRF	PD-8	3.43	350	336	231.88	DRF
TD-9	3.97	400	388	86.05	DRF	PD-9	3.38	368	330	110.80	DRF
TD-10	3.65	365	357	66.05	DRF	PD-10	3.32	350	324	290.88	DRF
TD-11	3.61	365	348	697.00	DRF	PD-11	5.98	600	584	13.00	DFF
TD-12	3.56	365	353	32.88	DRF	PD-12	5.80	580	567	11.17	DFF
TD-13	3.50	350	342	99.82	DRF	PD-13	5.50	550	538	15.57	DFF
TD-14	3.48	350	340	72.62	DRF	PD-14	4.98	498	487	118.60	DFF
TD-15	5.39	550	527	25.33	DFF ^2^	PD-15	3.86	400	378	698.00	DFF
TD-16	5.37	480	470	88.5	DFF	PD-16	3.62	368	354	690.00	DFF
TD-17	4.99	500	488	16.95	DFF						
TD-18	4.58	480	448	139.00	DFF						
TD-19	3.50	350	342	203.62	DFF						

^1^ Defect-related failure. ^2^ Defect-free failure.

**Table 6 materials-16-05099-t006:** Comparation of fatigue strength for the LAM samples in the four different groups.

Group	PD, DFF	TD, DFF	TD, DRF	PD, DRF
Fatigue strength at 10^5^ cycles, MPa	582.9	553.2	482.0	427.9
Fatigue strength at 10^6^ cycles, MPa	470.9	434.2	381.0	364.2

## Data Availability

The data presented in this study are available on request from the corresponding author.

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
