# Peer review of "An Investigation of the Anisotropic Fatigue Properties of Laser Additively Manufactured Ti-6Al-4V under Vibration Loading"

_materials, 2023, doi:10.3390/ma16145099_

Round 1
Reviewer 1 Report
In the presented work, vibration fatigue properties of LAM Ti-6Al-4V by laser melted deposition process was investigated. The title and abstract are satisfactory. I found the approach and conclusions to be robust and useful. The number of bibliographic references is sufficient. The state-of-the-art review presented in the Introduction part is comprehensive and follows a good logical structure. The testing guidelines and equipment used for carrying out the experiments are fully provided, and the obtained results are thoroughly presented and discussed accordingly. Finally, the Conclusions part does a good job in wrapping up the paper by summarizing the main findings. However, here are some comments and suggestions which can help improve the quality of the manuscript.
· The novelty of the work should be highlighted in the manuscript.
· In AMed parts, the surface roughness/defects influence the fatigue life. Was such condition taken into account in the present work?
· The details of FEM model should be included in the manuscript.
· Can the voids be reduced? It would be interesting to note how reduction in voids help in reducing the fracture initiation?
· The author is requested to do a thorough proofreading to improve the quality of the manuscript
· Better quality microstructure image indicating different phases can be included.
The author is requested to do a thorough proofreading to improve the quality of the manuscript
Author Response
Comments from reviewer #1:
Point 1: The novelty of the work should be highlighted in the manuscript.
Response:
Thank you for providing improvement. To highlight the novelty of this work, we have made some supplement to the conclusions.
Line 467 in the revised manuscript, we have supplemented as “The difference in fatigue performance can be totally distinguished by the different failure mechanisms shown to be present in the LAM material, i.e. process-induced defects and α colonies.”
Point 2: In AMed parts, the surface roughness/defects influence the fatigue life. Was such condition taken into account in the present work?
Response:
Yes, we have taken into account the influence of surface roughness and defects on the fatigue life in the present work.
The surface finish has a vital influence on the fatigue properties of AM alloys. In this work, to minimize the influence of surface roughness on fatigue behavior, the gauge section surfaces of all the samples were sanded with #180 – #5000 sandpaper and polished with diamond suspension. The measured Ra values ranged from 0.4 to 0.6 μm, indicating a relatively small variation in surface roughness. In our previous article [1], the surface treatment method is the same as what in this work, which avoids the influence of roughness.
In Section 3, the influence of process-induced defects on fatigue lives and fracture position of the LAM samples are analyzed. In Section 4.1, the influence of defects on the anisotropic fatigue behavior are discussed.
References:
[1] Zhao, S.; Yuan, K.; Guo, W.; He, Y.; Xu, Y.; Lin, X. A comparative study of laser metal deposited and forged Ti-6Al-4V alloy: Uniaxial mechanical response and vibration fatigue properties. Int. J. Fatigue 2019, 136, 105629.
Point 3: The details of FEM model should be included in the manuscript.
Response:
Thank you for providing improvement. In our work, the FEM model is used for modal analysis for cantilever sample in ABAQUS (Version 6.14). The modal analysis aims to assess the feasibility of the sample and provide reference data for frequency regulation and vibration stress calibration during experiments. The material parameters, model parameters and boundary conditions has been included in the manuscript. Based on these, more description of the FEM model has been given in Line 223 of the revised manuscript as follows.
As the mesh sensitivity affects the magnitude and distribution of the stress state near notch root, the mesh was refined at the notch region of the sample. The mesh size at the notch region is about 50 μm. The notch region consists of 6 element layers along the thickness of the sample. As a result, there are 124,676 elements and 198,376 nodes included in the model, as listed in Table 4. In addition, we ignored the anisotropy of material property in the model. This is because the purpose of the modal analysis was only to provide reference data for the frequency and vibration stress during experiments.
In order to clearly indicate details of the meshing, we have made changes to Figure 6a. The revised Figure 6 is shown as follows.
Figure 6. Modal analysis of Ti-6Al-4V sample by ABAQUS: (a) detail of the meshing, (b) displacement distribution, (c) stress distribution, and (d) stress distribution curve along the central path at different tip displacement. (e) Fitting curve for the maximum stress amplitude and sample tip displacement.
Point 4: Can the voids be reduced? It would be interesting to note how reduction in voids help in reducing the fracture initiation?
Response:
The voids can be greatly reduced through optimizing additive manufacturing process, but cannot be completely removed. In this work, the porosity of the LAM Ti-6Al-4V alloy is extremely low, approximately 0.006%. Due to the reduction in voids, the fatigue failure of some samples is caused by α colonies. The failure specimens caused by bunching have higher fatigue life. In such case, the difference in fatigue performance can be explained by different failure mechanisms, i.e. process-induced defects and α colonies.
Point 5: The author is requested to do a thorough proofreading to improve the quality of the manuscript
Response:
Thanks for your reminders. We have carefully checked this manuscript and revised the incorrect expressions.
Point 6: Better quality microstructure image indicating different phases can be included.
Response 6:
Thanks for your reminder. As you advised, we have replaced the inset in Figure 2c in the revised manuscript to indicate different phases (α phase & β phase). Meanwhile, we have supplemented the corresponding caption. The revised Figure 2 and caption is shown below.
Figure 2. (a) Three-dimensional metallography of the LAM Ti-6Al-4V. Microstructure in (b, c) deposition direction and (d, e) scanning direction. The white phase is α phase and the black phase is β phase.
Special thanks to you for your constructive comments and suggestions.

Reviewer 2 Report
Dear Authors,
the manuscript entitled "An investigation of anisotropic fatigue properties of laser additively manufactured Ti-6Al-4V under vibration loading" by Yan He and co-authors deals with the research on the additively manufactured Ti-6Al-4V samples. Materials produced using additive manufacturing technology are increasingly used in the construction of modern machines and are replacing traditional manufacturing technologies. Nevertheless, there is a need to verify the properties of materials under complex loading conditions as well as to develop an optimal method of its manufacture in terms of strength and durability of the structure. The paper is well justified, planned and written, and adds to the testing of the proporties of additively manufactured Ti-6Al-4V knowledge some new informations. I appreciate the contribution that the Authors made in experimental testing of printed parts as well as preparing the manuscript. However, in my opinion the manuscript needs to be improved in some fields and some general remarks as well as the specific comments are bellow.
Evaluation of the paper, general remarks, editorial comments/typos:
- The Abstract section should present quantitative results and not only the most important qualitative results and/or generic considerations. Significant improvements are expected in this section of the manuscript.
- line 48 and 49 - the Authors have written: "Accordingly, the microstructure exhibits strong directionality, and consequently result in anisotropy of mechanical properties [12–19].". This sentence refer to 8 articles at once. It makes sense to describe in detail these articles if they are important to the research presented by the Authors in the manuscript.
- the Authors present the goal of the research at the end of the Chapter 1 (Introduction). Please indicate in this section of the article what its novelty is in relation to the current state of knowledge.
- Research articles should present the directions of further research. I suggest adding one paragraph in the 5. Conclusion chapter.
- The above modifications should be implemented before considering the manuscript for publication. I hope these suggestions can help to improve the quality of this paper.
I wish you all the best.
Author Response
Comments from reviewer #2:
Point 1: The Abstract section should present quantitative results and not only the most important qualitative results and/or generic considerations. Significant improvements are expected in this section of the manuscript.
Response:
Thank you for providing improvement. The quantified results can explicitly reveal the differences of vibration fatigue properties between the four groups of samples. In the revised manuscript, we have supplemented the quantitative results about the anisotropic fatigue data.
Line 9: The Abstract section has been revised as “Laser additively manufactured (LAM) Ti–6Al–4V alloy has huge application potential in aero-space structural parts such as turbine blades. However, there are few studies on the fatigue properties of such LAM parts under vibration loading, particularly with regard to anisotropy. In this paper, vibration fatigue properties of LAM Ti-6Al-4V by laser melted deposition process were investigated along the transversely deposited (TD) direction and parallelly deposited (PD) direction. Through the first-order bending vibration experiments, LAM Ti-6Al-4V alloy exhibits obvious anisotropic fatigue properties and significant dispersion in fracture position. The fatigue strength at 106 cycles of the samples with defect-free failure features (DFF) at initiation sites is 470.9 MPa in PD and 434.2 MPa in TD, while those with defect-related failure features (DRF) at initiation sites is 364.2 MPa in PD and 381.0 MPa in TD. For the DFF group, the fatigue behavior is controlled by the prior β columnar grains with preferential orientation, which leads to enhanced fatigue crack propagation resistance for the PD samples. For the DRF group, having lower fatigue lives, the fatigue anisotropy strongly depends on the projection area of the lack-of-fusion defects relative to the loading direction, resulting in better fatigue performance for the TD samples.”
Meanwhile, we have added Table 6 in the revised manuscript to compare the fatigue strength for the LAM samples under the four different groups.
Table 6. Comparation of fatigue strength for the LAM samples under the four different groups.
|
Group |
PD, DFF |
TD, DFF |
TD, DRF |
PD, DRF |
|
Fatigue strength at 105 cycles, MPa |
582.9 |
553.2 |
482.0 |
427.9 |
|
Fatigue strength at 106 cycles, MPa |
470.9 |
434.2 |
381.0 |
364.2 |
The specific description for the quantitative results about the anisotropic fatigue data has been given in Line 372 of the revised manuscript as follows.
In order to quantify and compare the fatigue strength of samples under the four different groups, the calculated fatigue strength based on the linear fit are listed in Table 6. It can be seen that the samples of the DFF group in PD possessed a higher fatigue strength (470.9 MPa) at 106 cycles as compared to those in TD (434.2 MPa). The samples of the DRF group in PD possessed a lower fatigue strength (364.2 MPa) as compared to those in TD (381.0 MPa).
Point 2: Line 48 and 49 - the Authors have written: "Accordingly, the microstructure exhibits strong directionality, and consequently result in anisotropy of mechanical properties [12–19].". This sentence refers to 8 articles at once. It makes sense to describe in detail these articles if they are important to the research presented by the Authors in the manuscript.
Response:
Thanks for your suggestions. The 8 articles you mentioned has provided reference for the authors’ research, so we have described them in detail in the revised manuscript.
Line 51: We have revised as “Accordingly, the microstructure exhibits strong directionality, and consequently result in anisotropy of mechanical properties [12–14]. Lu et al. [15] and Carroll et al. [16] studied tensile mechanical anisotropy of LAM Ti-6Al-4V, and the results showed that samples along deposition direction present a better ductility, while samples along scanning direction exhibit a higher strength. Scholars suggested that the tensile anisotropy was caused by different deformation behaviors of the LoF defects, and was aggravated by the different deformation mechanism of the columnar grain boundaries [17,18]. Zhang et al. [19] studied the anisotropic tensile properties of selective electron beam melted (SEBM) Ti-6Al-4V and found the horizontally oriented samples had a lower yield strength, and a higher ductility. The anisotropic mechanical properties are affected by the process-induced defects and the epitaxially elongated grains.”
Meanwhile, the corresponding references ([13-19]) in the Reference section has also been adjusted in the revised manuscript.
Point 3: The Authors present the goal of the research at the end of the Chapter 1 (Introduction). Please indicate in this section of the article what its novelty is in relation to the current state of knowledge.
Response:
Thanks for your suggestion. We have carefully checked the Introduction section and have supplemented one sentence to clearly indicate the novelty in relation to the current state of knowledge.
Line 75 in the revised manuscript, we have supplemented as “However, traditional fatigue experiments do not consider the natural frequency of the samples.”
Point 4: Research articles should present the directions of further research. I suggest adding one paragraph in the 5. Conclusion chapter.
Response:
Thank you for providing improvement. Following your advice, we have added the directions of further research in the last paragraph of the Conclusion section in the revised manuscript as follows.
This study proves that the initiation and propagation of vibration fatigue cracks in AM materials are closely related to the process-induced defects and microstructure (columnar grains or α colonies). The results and discussion provide a basis for further regulating the vibration fatigue lives of AM materials. For AM, anisotropy contribution to fatigue performance needs to be better understood as well as factors not considered here such as surface finish and porosity spatial distribution. This points out the directions for the authors’ future research.
Since the description of future research directions has been added to the Conclusion section, we have revised the chapter title in the revised manuscript.
Line 463: We changed the title to “Conclusions and future outlook”.
Special thanks to you for your constructive comments and suggestions.
